# Effect of Residual Trace Amounts of Fe and Al in Li[Ni_1/3_Mn_1/3_Co_1/3_]O_2_ Cathode Active Material for the Sustainable Recycling of Lithium-Ion Batteries

**DOI:** 10.3390/ma14092464

**Published:** 2021-05-10

**Authors:** Seongdeock Jeong, Sanghyuk Park, Mincheol Beak, Jangho Park, Jeong-Soo Sohn, Kyungjung Kwon

**Affiliations:** 1Department of Energy & Mineral Resources Engineering, Sejong University, Seoul 05006, Korea; ektmfrhw@gmail.com (S.J.); shpark@sejong.ac.kr (S.P.); doqmsalscjf@naver.com (M.B.); wkdgh1229@naver.com (J.P.); 2Mineral Resources Research Division, Resources Recovery Research Center, Korea Institute of Geoscience and Mineral Resources, Daejeon 34132, Korea

**Keywords:** lithium-ion battery, Li[Ni_1/3_Mn_1/3_Co_1/3_]O_2_, aluminum, iron, resynthesis

## Abstract

As the explosive growth of the electric vehicle market leads to an increase in spent lithium-ion batteries (LIBs), the disposal of LIBs has also made headlines. In this study, we synthesized the cathode active materials Li[Ni_1/3_Mn_1/3_Co_1/3_]O_2_ (NMC) and Li[Ni_1/3_Mn_1/3_Co_1/3_Fe_0.0005_Al_0.0005_]O_2_ (NMCFA) via hydroxide co-precipitation and calcination processes, which simulate the resynthesis of NMC in leachate containing trace amounts of iron and aluminum from spent LIBs. The effects of iron and aluminum on the physicochemical and electrochemical properties were investigated and compared with NMC. Trace amounts of iron and aluminum do not affect the morphology, the formation of O3-type layered structures, or the redox peak. On the other hand, the rate capability of NMCFA shows high discharge capacities at 7 C (110 mAh g^−1^) and 10 C (74 mAh g^−1^), comparable to the values for NMC at 5 C (111 mAh g^−1^) and 7 C (79 mAh g^−1^), respectively, due to the widened interslab thickness of NMCFA which facilitates the movement of lithium ions in a 2D channel. Therefore, iron and aluminum, which are usually considered as impurities in the recycling of LIBs, could be used as doping elements for enhancing the electrochemical performance of resynthesized cathode active materials.

## 1. Introduction

Lithium-ion batteries (LIBs) are suitable for portable IT devices, energy storage systems, and electric vehicles (EVs) due to their excellent electrochemical performance, related to their long cycling life and high energy density [1]. In particular, the explosive growth of the EV market has led to an increase in the production of LIBs [2]. The development of the EV industry has inevitably led to the generation of large amounts of spent LIBs [3]. Because there are many valuable metals in cathode active materials from the spent LIBs, they must be disposed of safely and properly due to their hazardous effects on human health and the environment [4].

Spent LIBs contain considerable impurities such as iron, aluminum, and copper, in addition to the valuable metals included in the cathode active materials, such as lithium, nickel, and cobalt. Moreover, trace impurities such as zinc, magnesium, calcium, and sodium can emerge during the LIB recycling process [5]. In particular, it is important to recover cobalt and lithium from an economic viewpoint because of their resource scarcity and cost [6]. The recycling process for the spent LIBs is generally comprised of physical and chemical processes. The physical processes include a mechanical separation process, a dissolution process, and a thermal treatment, while the chemical processes include pyrometallurgy, biometallurgy, and hydrometallurgy. In particular, an acidic leaching process belonging to hydrometallurgy is mainly used to dissolve the metals because of its simple working conditions, low energy consumption, low dust generation, and high recovery rate [7]. The obtained leachate goes through a series of chemical processes, such as solvent extraction, chemical precipitation, and an electrochemical process for purifying the metals selectively [8].

The direct regeneration of cathode active materials via a co-precipitation method using the leachate has been considered as a promising recycling strategy. Although the leachate goes through a combination of several recycling processes as impurity removal steps, it is difficult to remove the impurities completely [9]. Accordingly, the trace amount of impurities that are not fully eliminated from the leachate are incorporated during the co-precipitation for the direct regeneration of cathode active materials and affect the electrochemical performance of the resynthesized cathode active materials. In our previous study, the effects of impurity elements including iron, aluminum, tin, lithium, copper, and sodium were investigated individually at impurity levels [10,11,12,13,14,15,16]. It was reported that Li[Ni_1/3_Mn_1/3_Co_1/3_]O_2_ (NMC) containing trace amounts of Fe or Al via co-precipitation showed better LIB performance in terms of cyclability and rate capability compared to a pristine sample [10,11,12]. In the case of iron, NMC structures with iron contents of 0.05 and 0.25 mol% exhibited an apparently improved rate capability at high C-rates with an increased initial capacity, which is attributed to their expanded lattice volume compared to a virgin sample [11,17]. In the meantime, the effects of trace amounts of aluminum were also investigated for NMC [10]. The NMC structure with an aluminum content of 0.05 mol% showed an improved rate capability at high C-rates. However, excess iron and aluminum contents above 1 mol% lowered the electrochemical performance, which was induced by the severe cation disordering in the case of iron and a detrimental influence on the growth of secondary particles during the co-precipitation reaction for aluminum [10,11]. Considering the positive effects of the trace amounts of iron and aluminum incorporated in the NMC structure, the influence of multiple ions needs to be studied further, which is closer to the actual leachate system.

In this study, we synthesized iron and aluminum co-doped NMC (NMCFA) cathode active materials via the co-precipitation method, which simulates the resynthesis of NMC in leachate containing trace amounts of iron and aluminum from spent LIBs. A tolerance level of 0.05 mol% for iron and aluminum was introduced during the co-precipitation, aiming at the synergetic effect of enhancing the structural and electrochemical properties. A series of electrochemical analyses including the galvanostatic intermittent titration technique (GITT), cyclic voltammetry (CV), and electrochemical impedance spectroscopy (EIS) were systematically conducted to elucidate the effects of co-doping process.

## 2. Experimental Section

### 2.1. Synthesis of Materials

Hydroxide precursors, Ni_1/3_Mn_1/3_Co_1/3_(OH)_2_ and Ni_1/3_Mn_1/3_Co_1/3_Fe_0.0005_Al_0.0005_(OH)_2_, were synthesized using a co-precipitation method. An amount of 1 M of NH_4_OH solution was used as a chelating agent and 2 M of metal solution (NiSO_4_·6H_2_O, MnSO_4_·H_2_O, and CoSO_4_·7H_2_O in a molar ratio of 1:1:1) with controlled amounts of FeSO_4_·H_2_O and Al_2_(SO_4_)_3_·16H_2_O was pumped into a continuous stirred tank reactor (CSTR, 5 L capacity). An amount of 2 M of NaOH solution was automatically injected into the CSTR to maintain the desired pH of 10.5 during the co-precipitation reaction. The reaction was kept at a temperature of 50 °C and a stirring speed of 1000 rpm under a nitrogen atmosphere. Prepared precursors were sampled after about 60 h to obtain the proper size distribution of secondary particles. The obtained precursor powder was filtered and thoroughly washed with distilled water, and then it was dried in an oven overnight at 60 °C. The dried precursors were mixed with LiOH·H_2_O in a molar ratio of 1:1.05 and calcined at 1000 °C for 8 h to synthesize Li[Ni_1/3_Mn_1/3_Co_1/3_]O_2_ and Li[Ni_1/3_Mn_1/3_Co_1/3_Fe_0.0005_Al_0.0005_]O_2_, which are referred to as NMC and NMCFA, respectively.

### 2.2. Physciochemical Analysis

The morphology and elemental distribution of the synthesized cathode active materials were measured by a field emission scanning electron microscope (FE-SEM, SU-8010, HITACHI Ltd., Japan) and energy dispersive spectroscopy (EDS), respectively. The content of metals in the cathode active materials was analyzed by inductively coupled plasma-mass spectrometry (ICP-MS). The crystal structure and corresponding lattice parameters of samples were investigated by X-ray diffraction (XRD, D/Max-2500, Rigaku Ltd., Japan) with Cu Kα radiation (λ = 1.5406 Å) in the 2θ range of 10–80° at a step size of 0.02° and the FullProf software. The specific surface area was determined by Brunauer–Emmett–Teller (BET, BELSORP-max, BEL Inc., USA) analysis to calculate the lithium-ion diffusion coefficient of the samples.

### 2.3. Electrochemical Analysis

Cathodes with a loading level ranging from 10 to 11 mg cm^−2^ consisted of the cathode active materials (NMC and NMCFA), carbon black (Super P) as a conducting agent, and polyvinylidene fluoride (KF 1100) as a binder in a weight ratio of 95:2:3. For a cathode slurry, an N-methyl-pyrrolidinone solution of 59 wt.% total solid content was mixed thoroughly and casted on an Al current collector. The casted cathode slurry was dried in an oven at 80 °C for 2 h, followed by roll-pressing to a desired thickness. After that, electrochemical tests were carried out using CR2032-type coin cells with a battery cycler system (WBCS3000L, WonAtech Ltd., Korea). Li metal foil as an anode, a polyethylene film as a separator, and 1 M LiPF_6_ in ethylene carbonate/ethylmethyl carbonate mixture with a volume ratio of 1:2 as an electrolyte were used for the cell assembly. LIB performances were measured in a voltage range from 3.0 to 4.3 V at 25 °C, and 150 mAh g^−1^ was set asa nominal capacity of 1 C. The first two cycling at 0.1 C was taken as a formation step to activate the cells. All electrochemical potentials were based on Li/Li^+^ unless mentioned otherwise. Rate capability tests were performed at various discharge C-rates from 0.1 to 10 C with a fixed charge C-rate of 0.2 C, while cycling performance was measured at 1 C for 50 cycles. CV measurements were conducted in a potential range of 3.0–4.3 V at a scan rate of 0.1 mV s^−1^. EIS was conducted in a frequency range from 1000 kHz to 1 mHz with an alternating voltage amplitude of 15 mV. GITT was carried out to obtain Li ion diffusion coefficient constants, where cells were charged for 20 min and rested for 20 min up to 4.3 V with a constant current of 150 mA g^−1^ (1 C).

## 3. Results and Discussion

### 3.1. Physicochemical Charateristics

SEM images of the synthesized cathode active materials at different magnifications are shown in Figure 1. Both NMC and NMCFA have spherical secondary particles consisting of granular primary particles, which indicates that trace amounts of iron and aluminum do not affect the morphological properties of precursors. This result is consistent with the previous literature that has investigated the effects of iron or aluminum on the morphology of NMC [10,11]. Table 1 shows the concentration of constituent metals in the cathode active materials analyzed by ICP-MS, which indicates that both samples were synthesized in accordance with the designed composition. In addition, Figure 2 shows the even elemental distribution of constituent metals (Ni, Mn, Co, Fe, and Al) in the cathode active materials of NMC and NMCFA.

Powder XRD patterns of NMC and NMCFA are presented in Figure 3. The XRD peaks of the cathode active materials correspond to a layered structure based on the hexagonal α-NaFeO_2_ with a space group of R-3m. The (006)/(102) and (108)/(110) peaks show clear peak splitting, indicating a well-ordered layered structure. It is known that if there is no distortion in the direction of the c-axis in a hexagonal lattice or the crystal structure is perfect cubic, the peak splitting of (006)/(102) and (108)/(110) peaks is not observed. However, the oxygen sublattice in the typical O3-type (ABCABC stacking) layered transition metal oxide structure forms clear splitting peaks in the XRD patterns [18]. Consequently, trace amounts of iron and aluminum do not impede the formation of an O3-type layered structure, consistent with the previous literature [10,11]. Lattice parameters of the cathode active materials based on XRD data are calculated by Rietveld refinement as shown in Table 2. The refined lattice parameters of the NMC sample are in good agreement with the values reported in the previous literature [19]. In the O3-type layered structure, lithium ions, transition metal ions, and oxygen ions are generally located on 3a, 3b, and 6c sites, respectively, as displayed in Figure 3b. Some Ni^2+^ ions on 3b sites can migrate to 3a sites during the synthesis process because the ionic radii of Ni^2+^ and Li^+^ are similar, which is called cation mixing [20,21]. However, the calculated values of intensity ratio of (003) to (104) peaks for both NMC and NMCFA are over 1.2, which would mean that there is less cation disordering. Meanwhile, no obvious differences in the lattice parameters a and c are observed between NMC and NMCFA, as seen in Table 2. The layer thicknesses of the transition metal (T_TM_) and the interslab (T_Li_) were calculated to further investigate the structural properties of NMC and NMCFA in detail. In the case of NMCFA, the T_TM_ was reduced, whereas the T_Li_ was increased compared to that of NMC. The increased T_Li_ of NMCFA could lead to a kinetically favorable effect for an electrochemical property related to the diffusion of lithium ions.

### 3.2. Electrochemical Characteristics

Figure 4a shows the initial charge/discharge curves of the cathode active materials at 0.1 C in a voltage range between 3.0 and 4.3 V at 25 °C. While both samples show similar behavior during the charging and discharging processes, NMC has higher charge/discharge capacities than NMCFA including the electrochemically inactive elements of iron and aluminum. The initial CV curves of the samples in Figure 4b also describe the slightly decreased current of NMCFA compared to NMC, with no distinct redox peaks contributing to a capacity that is induced by iron and aluminum species, which are electrochemically inactive in the potential range of 3.0–4.3 V. An identical major redox couple at 3.86 V for the charging process and 3.64 V for the discharging process, which was also observed in the previous report [22], emerges for both NMC and NMCFA. It is well known that nickel ions with a dominant oxidation state of +2 in the NMC structure can participate in electrochemical redox reactions between Ni^2+^/Ni^4+^ during the charge and discharge processes [23]. To investigate the degree of overpotential of NMCFA compared to NMC with an increase in the C-rate, a rate capability test was carried out at various discharge C-rates from 0.1 to 5 C, with a fixed charge C-rate of 0.2 C in Figure 4c. In contrast to the discharge process at the low C-rate of 0.1 C in Figure 4a, which shows a similar overpotential for both NMC and NMCFA, NMCFA clearly exhibits a relatively reduced overpotential compared to NMC at the higher C-rates than 1 C, which results in the higher discharge capacity of NMCFA. The reason for the better performance at high C-rates is attributed to the widened interslab thickness of NMCFA, as mentioned above in Figure 3.

In order to scrutinize the improved rate performance of NMCFA at the high C-rates, a rate capability test under different conditions from Figure 4c was conducted, where the C-rate was increased from 0.1 to 10 C step-wise with 3 cycles at each C-rate. The discharge capacity and the corresponding relative rate capability of NMC and NMCFA are plotted in Figure 5. There was no obvious capacity fading when the discharge C-rate returned to 0.1 C after 10 C. The discharge capacities of NMCFA displayed lower values than those of NMC at low C-rates up to 1 C. However, the capacities of NMCFA exceeded those of NMC at high C-rates over 1 C, and the difference in discharge capacity increased with the increase in the C-rate. It is noted that the discharge capacities of NMCFA at 7 C (110 mAh g^−1^) and 10 C (74 mAh g^−1^) are comparable to the values for NMC at 5 C (111 mAh g^−1^) and 7 C (79 mAh g^−1^), respectively. It can be concluded that NMCFA has the advantage of the facile migration of the lithium ion compared to NMC at a high C-rate condition, which requires rapid lithium ion diffusion. Moreover, NMCFA shows a superior relative rate capability, which is the capacity retention at different C-rates relative to the discharge capacity at 0.1 C, to NMC at all C-rates, as displayed in Figure 5b. The rate capability enhancement of NMCFA could be attributed to its increased T_Li_ compared to NMC, as shown in Table 2.

In general, the less cation mixing and widened interslab thickness make it easy to move lithium ions in 2D channels [24]. Therefore, it is concluded that the co-doping of iron and aluminum to NMC leads to an improvement in the rate capability due to the widened interslab and less cation mixing, which were confirmed in Figure 3. We further conducted electrochemical impedance spectroscopy (EIS) and the galvanostatic intermittent titration technique (GITT) to elucidate the improved rate performance of NMCFA.

Figure 6 shows the Nyquist plot of NMC and NMCFA before and after the rate test, which is displayed in Figure 5. EIS was measured on the samples at the charged state, and an equivalent circuit consisting of AC impedance elements, which represent the internal resistances of the cells, is given as an inset in Figure 6. Solution resistance (R_s_) from electrolyte and surface film resistance (R_sf_) from solid electrolyte interphase (SEI) after the formation step, charge transfer resistance (R_ct_), and Warburg impedance (W_o_) related to the diffusion of lithium ions are included in the equivalent circuit, where the distinguishable slope of Warburg impedance at a low-frequency region originates from the extraction of lithium ions occurring at a suitable potential [25]. The cell capacitances (C_sf_ and C_dl_) of electrochemical reactions could be expressed as constant-phase elements (CPE) for the film capacitance of the SEI layer as CPE1 and double-layer capacitance of the cathode as CPE2, respectively [26]. In general, EIS data consist of the first semicircle at a high-frequency region related to lithium-ion diffusion through the SEI layer, the second semicircle at a mid-frequency region related to a charge transfer reaction between the electrode and the electrolyte, and the slope at a low-frequency related to lithium-ion diffusion in bulk materials [27,28]. The feature of increased R_s_ in the case of NMC compared to NMCFA after the rate test suggests that the depletion of the electrolyte and the formation of a microcrack in the secondary particles of NMC are increasingly generated during the test [29]. Although the total cell resistance of NMC after the rate test is too ambiguous to be divided into the two semicircles, as explained above (see Figure 6a), it is clear that the smaller overall resistance of NMCFA including R_sf_ and R_ct_ is maintained after the test. Thus, it can be explained that the superior rate performance of NMCFA stems from a lower internal resistance than that of NMC.

GITT was carried out to estimate the lithium-ion diffusion coefficient (*D_Li_^+^*) and to further understand the kinetic behavior of the cathode active materials as presented in Figure 7. The experimental GITT is based on chronopotentiometry under equilibrium conditions and is used as an effective measurement to obtain the *D_Li_*^+^ [30]. The GITT curves of NMC and NMCFA during the second charge process of the formation step are displayed in Figure 7a. The current pulse vs. voltage profile for a single titration is presented in Figure 7b, where ∆*E_τ_* is the total change of cell voltage during the current flux and ∆E_s_ is the voltage change between steady states [31]. The *D_Li_^+^* can be determined by solving Fick’s second law of diffusion as follows [32]:(1)$$D_{Li^{+}}=\frac{4}{\pi }{\left(\frac{mV}{MS}\right)}^{2}{\left(\frac{\Delta E_{S}}{\tau \left(\frac{dE}{d\sqrt{\tau }}\right)}\right)}^{2}\;\left(\tau \ll \frac{L^{2}}{D_{Li^{+}}}\right),$$
where *m* (g) is the mass, *V* (cm^3^ mol^−1^) is the molar volume, and *M* (g mol^−1^) is the molecular weight of the cathode active materials. *S* (cm^2^) is the contact area between the electrode and the electrolyte, and *L* (cm) is the lithium-ion diffusion length. Equation (1) can be further simplified into Equation (2) if *E* vs. $\sqrt{\tau }$ shows a straight-line behavior during the titration step, as seen in Figure 7c [33,34].
(2)$$D_{Li^{+}}=\frac{4}{\pi \tau }{\left(\frac{mV}{MS}\right)}^{2}{\left(\frac{\Delta E_{S}}{\Delta E_{\tau }}\right)}^{2}\left(\tau \ll \frac{L^{2}}{D_{Li^{+}}}\right).$$

Figure 7d exhibits the calculated *D_Li_^+^* using Equation (2) as a function of cell voltage. The calculated *D_Li_^+^* values of NMCFA ranging from 6.0 × 10^−13^ to 13.0 × 10^−12^ cm^2^ s^−1^ are comparatively higher than those of NMC in the entire voltage range. Therefore, NMCFA has better kinetic behavior for the (de)intercalation of lithium ions than NMC due to the higher diffusion coefficient, regardless of the C-rate condition.

Figure 8 shows the cycling performance of NMC and NMCFA at 1 C in the potential range of 3.0–4.3 V. NMCFA has a relatively reduced discharge capacity compared to NMC due to the electrochemically inactive iron and aluminum as confirmed in the CV test in Figure 4b. An abrupt decay in capacity occurred after 45 cycles in the case of NMCFA, with NMC still showing a gradual deterioration for up to 50 cycles. Given that a trace amount of iron or aluminum-doped NMC shows similar or better cyclability than the pristine NMC sample under the same experimental conditions [10,11], it could be concluded that the synergistic effect of iron and aluminum doping to NMC on the cyclic performance is not achieved. Overall, the multiple ion (iron and aluminum)-doping strategy into NMC is effective for the rate capability enhancement, but it does not work for the cyclic performance.

## 4. Conclusions

Herein, we have demonstrated the high rate performance of layered Li[Ni_1/3_Mn_1/3_Co_1/3_]O_2_ prepared by co-precipitation with trace amounts of iron and aluminum impurity as a direct regeneration method. The obtained cathode active materials of NMC and NMCFA exhibited spherical secondary particles consisting of granular primary particles and layered structures, with a space group of R-3m. Whilst there were no specific differences in the morphology and the element distribution between NMC and NMCFA, the T_TM_ did reduce and the T_Li_ did increase compared to those of NMC in the case of NMCFA. We expected that the increased T_Li_ could lead to a kinetically favorable effect for an electrochemical property related to the diffusion of lithium ions, and as a result, the rate capability of NMCFA showed higher discharge capacities at 7 C (110 mAh g^−1^) and 10 C (74 mAh g^−1^), comparable to the values for NMC at 5 C (111 mAh g^−1^) and 7 C (79 mAh g^−1^), respectively. The high discharge capacities at high C-rates of NMCFA were supported by EIS and GITT, which displayed modest increases in resistances and the high lithium-ion diffusion coefficient, respectively. Therefore, it can be concluded that the widened interslab thickness of NMCFA has the advantage of the facile migration of the lithium ion to move into 2D channels, which requires rapid lithium-ion diffusion. 

## Figures and Tables

**Figure 1 materials-14-02464-f001:**
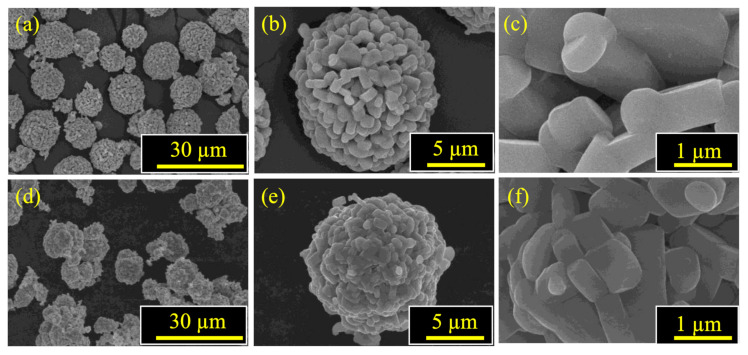
SEM images of the synthesized cathode active materials: (**a**–**c**) NMC and (**d**–**f**) NMCFA.

**Figure 2 materials-14-02464-f002:**
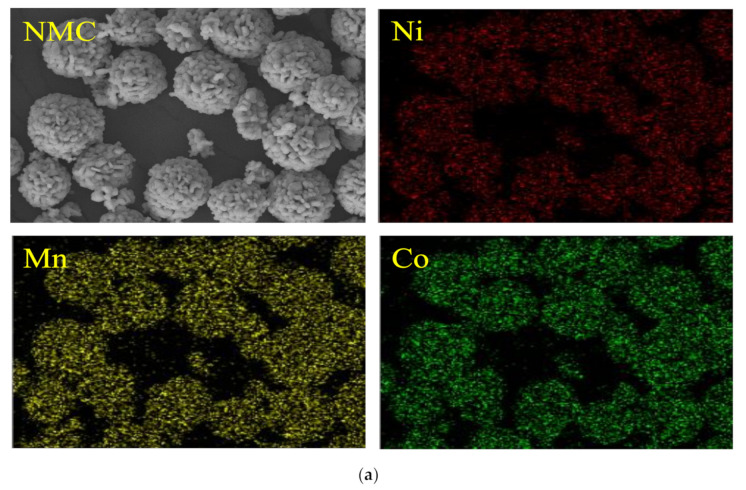
Elemental distribution of the cathode active materials analyzed by EDS: (**a**) NMC and (**b**) NMCFA.

**Figure 3 materials-14-02464-f003:**
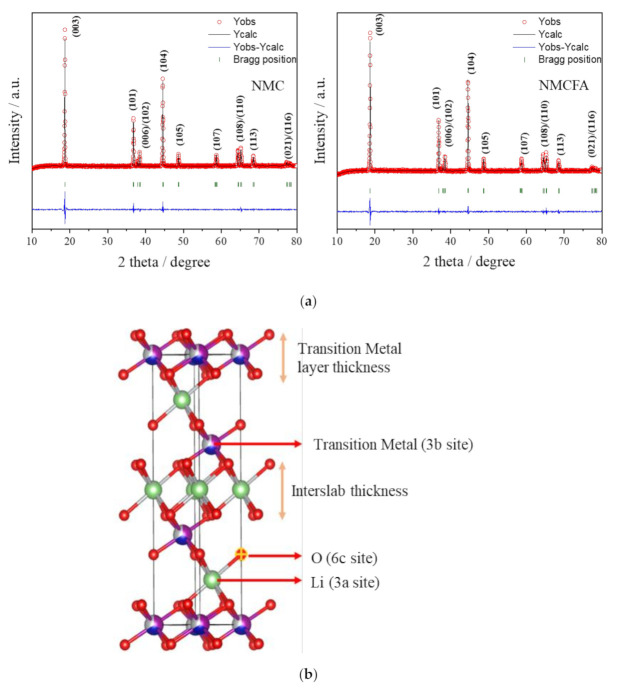
(**a**) XRD patterns of NMC and NMCFA analyzed by Rietveld refinement (NMC: R_p_ = 16.2, R_wp_ = 13.9, and χ^2^ = 6.66/NMCFA: R_p_ = 20.3, R_wp_ = 13.7, and χ^2^ = 3.35). (**b**) Schematic illustration of a hexagonal lattice.

**Figure 4 materials-14-02464-f004:**
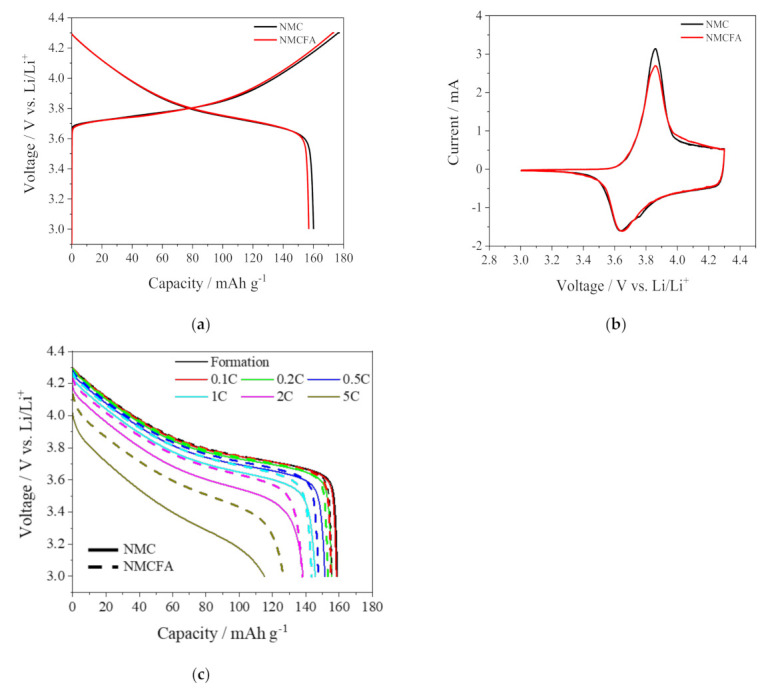
Comparison of the galvanostatic charge and discharge processes of NMC and NMCFA: (**a**) voltage profiles of initial cycling at 0.1 C, (**b**) initial cyclic voltammograms at a scan rate of 0.1 mV s^−1^, and (**c**) discharge curves at different C-rates from 0.1 to 5 C.

**Figure 5 materials-14-02464-f005:**
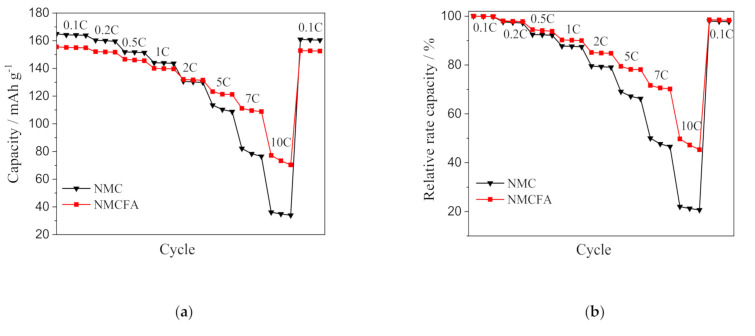
(**a**) Rate performance and (**b**) the corresponding relative rate capability of NMC and NMCFA at various C-rates.

**Figure 6 materials-14-02464-f006:**
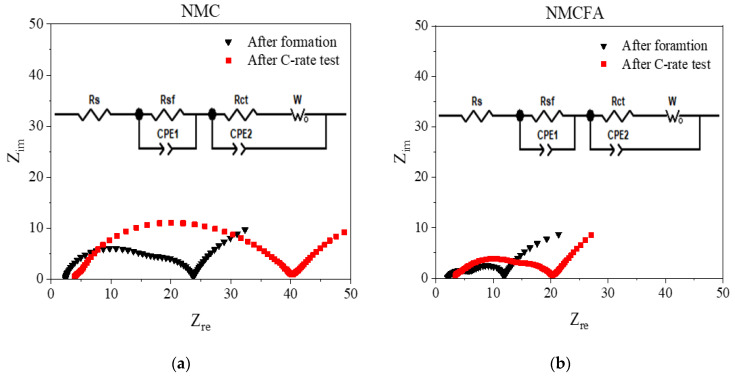
Nyquist plots of (**a**) NMC and (**b**) NMCFA before and after the rate capability test with insets of equivalent circuits.

**Figure 7 materials-14-02464-f007:**
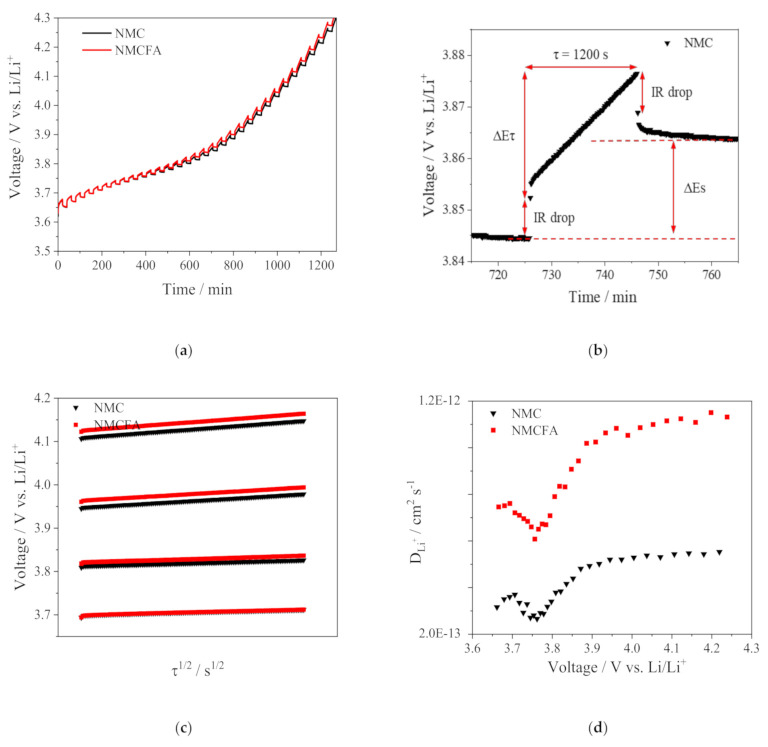
(**a**) GITT curves of NMC and NMCFA during the charge process at a current density of 0.1 C in the voltage range between 3.0 and 4.3 V. (**b**) GITT curve at a single titration of NMC. (**c**) Linear behavior in the relationship of E vs. √τ over the whole voltage range. (**d**) Lithium-ion diffusion coefficients of NMC and NMCFA calculated from the GITT curves as a function of the cell voltage.

**Figure 8 materials-14-02464-f008:**
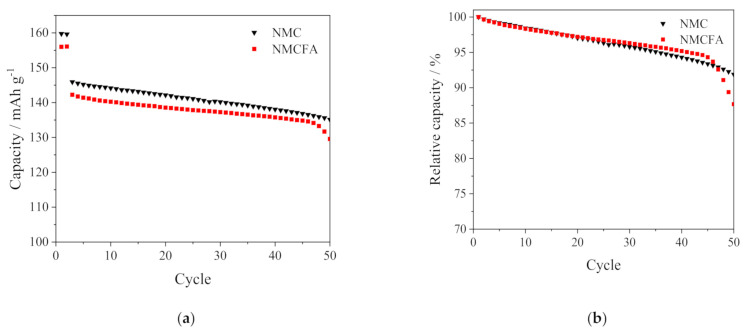
(**a**) Cycling performance and (**b**) capacity retention of the NMC and NMCFA at 1 C.

**Table 1 materials-14-02464-t001:** Elemental composition and relative standard deviation of the cathode active materials resulting from ICP-MS analysis.

Sample	Elemental Composition (mol %)					
	Ni	Mn	Co	Fe	Al	Total
NMC	32.53	33.66	33.81	N.D.	N.D.	100
NMCFA	32.21	32.65	35.03	0.056	0.054	100
%RSD	3.02	1.62	3.62	10.71	7.41	-

**Table 2 materials-14-02464-t002:** Crystallographic parameters and relative standard deviation of the cathode active materials.

Sample	a (Å)	c (Å)	Volume (Å^3^)	I(003)/I(104)	T_TM_ (Å)	T_Li_ (Å)
NMC	2.8578	14.2172	100.5554	1.3365	2.1165	2.6226
NMCFA	2.8547	14.2113	100.2957	1.3678	2.0843	2.6528
%RSD	0.14	0.049	0.36	-	-	-

## Data Availability

The data presented in this study are available on request from the corresponding author.

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
