# Peer review of "Effect of Residual Trace Amounts of Fe and Al in Li[Ni1/3Mn1/3Co1/3]O2 Cathode Active Material for the Sustainable Recycling of Lithium-Ion Batteries"

_materials, 2021, doi:10.3390/ma14092464_

Round 1

Reviewer 1 Report

The authors present here an interesting work on the effect of Fe and Al cations doping at trace level on NMC, simulating contamination from the recycling process. While the work is well presented, the following major corrections need to be addressed before the paper is in a state for acceptance in Materials:

- Considering the trace amount of the dopants, the authors must indicate the standard deviation of the results in their ICP and Rietveld refinement data. In addition, for the Rietvel refinement, the authors should indicate the quality factor of their refinement and include the fit data curve and the residuals in the graph.

- Figure 2 is incorrectly labelled as Figure 4.

- In line 130 the authors indicate a current for the GITT current of 150 mA/g as 0.1C, is this correct? Is the 20 mins rest time in the pulse enough to obtain the relaxed voltage value for the steady-state? Which particle size have the authors employed for the diffusion coefficient calculation? In most cases, referring to diffusion length could be more accurate.

- The title is misleading, the authors are not resynthesising NMC from spent batteries, this need to be modified in the title. 

- Have the authors verify the reproducibility of their test, especially the abrupt decay of cyclability of the NMCFC material after 45 cycles?

Reviewer 2 Report

Dear Author,

Journal Materials (ISSN 1996-1944)

Manuscript ID materials-1202153

Type Article

Number of Pages 13

Title Electrochemical properties of trace Fe and Al co-doped Li[Ni1/3Mn1/3Co1/3]O2 in the resynthesis of cathode material from spent Li-ion batteries

Authors Seongdeock Jeong , Sanghyuk Park , Mincheol Beak , Jangho Park , Jeong-Soo Sohn * , Kyungjung Kwon

In my opinion, the paper deals with an interesting subject, related to Li-rich cathode materials who attracted great interest due to the large enhancement of their electrochemical performances.

The novelty is noticeable. The authors already have several similar articles published in high-ranking journals, which shows the great experience in this field. The work was created as a fusion of two recently published and highly regarded articles.

General remarks:

  1. If the author in previous works determined the content limit for Fe I Al as impurities, why did he choose the content of iron only 0.05%? (up to 0.25%) Maybe the content of Fe must be higher to enhance the material conductivity or stabilize the crystal structure thus enhancing the cyclability of the active material.

Please find attached the revised manuscript.

Recommend to the author:

  1. The Abstract: No objections
  2. The Introduction: The introduction provides a good, generalized background of the topic that quickly gives the reader an appreciation of the wide range of applications for this technology.
  3. The Experimental and The Results and discussion: The experimental apparatus is quite standard, and is appropriate for the study. I don’t think any additional experiments are necessary to validate the results presented here.
    • EIS data are shortly described in the text.

4. The Conclusion: After corrections modify the conclusion if you feel you need it.

5. Literature: Check all reference. REF 31,32,33…check in the text

Suggestion for Authors

I advise the authors to carefully revise the manuscript in order to avoid errors and to clarify some points.

Please, consider all these comments.

Sincerely,

Reviewer 3 Report

This research work "Electrochemical properties of trace Fe and Al co-doped Li[Ni1/3Mn1/3Co1/3]O2 in the resynthesize of cathode material from spent Li-ion batteries" shows the addition of iron and aluminum during LIB recycling into the NMC cathode lattice structure could enhance the rate capability due to the a widened interslab layer distance. Overall, it is a nice work with comprehensive studies, and thus, I suggest it to be published with a minor revision, here are my comments:

Line 25, it states "a widened interslab layer thickness", would "interslab layer distance" be more appropriate?

Line 76, the title indicates the Fe and Al doping is from the spent LIB, however, the actual synthesis just simulates such a process. So the title may need to be corrected to avoid any misleading.

Line 193, I am not fully convinced that the better electrochemical performance is attributed to the widened interslab layer thickness. If so, doping metal ions could also lead to an improved electrochem. Would the Fe or Al ions themselves have any influence?

Line 242-250, unbold the text.

Line 273-274, usually the doping of Al into NMC cathode could increase the lattice structure stability and thus the cycling performance, why multiple ions (Fe and Al) doping does not work here? Please comment.

Round 2

Reviewer 1 Report

The authors have addressed the majority of the comments made by this reviewer, and acceptance of this paper is recommended after the authors also include the standard deviations for the concentrations of the dopants and the unit cell parameters.
